# Epidemiology of aerophagia in children and adolescents: A systematic review and meta-analysis

Shaman Rajindrajith [1,2]*, Damitha Gunawardane[3], Chandrani Kuruppu[4], Samath D. Dharmaratne[3], Nipul K. Gunawardena[5], Niranga M. Devanarayana[6]

**1** Department of Pediatrics, Faculty of Medicine, University of Colombo, Colombo, Sri Lanka, **2** The Lady Ridgeway Hospital for Children, Colombo, Sri Lanka, **3** Department of Community Medicine, Faculty of Medicine, University of Peradeniya, Peradeniya, Sri Lanka, **4** Medical Library, Faculty of Medicine, University of Colombo, Colombo, Sri Lanka, **5** Department of Parasitology, Faculty of Medicine, University of Kelaniya, Ragama, Sri Lanka, **6** Department of Physiology, Faculty of Medicine, University of Kelaniya, Ragama, Sri Lanka

* shamanrajindrajith4@gmail.com

## Abstract

### Background

Aerophagia is a common functional gastrointestinal disorder among children. The disease leads to symptoms related to air in the intestine leading to burping, abdominal distension, and excessive flatus. We aimed to perform a systematic review and a meta-analysis to assess the epidemiology of aerophagia in children.

### Methods

We conducted a thorough electronic databases (MEDLINE, EMBASE, PsycINFO and Web of Science) search for all epidemiological surveys conducted in children on aerophagia. All selected studies were assessed for their scientific quality and the extracted data were pooled to create a pooled prevalence of aerophagia.

### Results

The initial search identified 76 titles. After screening and in depth reviewing, 19 studies representing data from 21 countries with 40129 children and adolescents were included in the meta-analysis. All studies have used standard Rome definitions to diagnose aerophagia. The pooled prevalence of aerophagia was 3.66% (95% Confidence interval 2.44–5.12). There was significant heterogeneity between studies [$I^2$ 98.06% with 95% Confidence interval 97.70–98.37). There was no gender difference in prevalence of aerophagia in children. The pooled prevalence of aerophagia was highest in Asia (5.13%) compared to other geographical regions.

### Conclusion

In this systematic review and meta-analysis, we found aerophagia has a significant prevalence across the world.

**Data Availability Statement:** This is a systematic review. All data are in the public domain. The

characteristics of the studies and extracted data are presented in Table 1 of the paper.

**Funding:** The authors received no specific funding for this work.

**Competing interests:** The authors have declared that no competing interests exist.

## Introduction

Aerophagia (AP) denotes excessive swallowing of air and symptoms often accompanying it such as burping, increased flatus and abdominal distension. Although it seems to be a disease of insignificance, AP inflicts an undesirable effect on the lives of children. AP negatively affects health-related quality of life of affected children [1]. Sagawa and co-workers have reported that AP reduces the quality of school life, which possibly affect their future [2]. Furthermore, AP is also associated with psychological maladjustment and psychological stress [1, 3]. Other than symptoms due to air in the gastrointestinal tract, these children also suffer from a multitude of other somatic symptoms [1, 3]. AP, in its severest forms, is associated with intestinal perforation and volvulus [4, 5].

After the release of the Rome criteria, there had been a growing number of epidemiological surveys that report the prevalence of AP among school children across the world [6, 7]. However, a systematic review and a meta-analysis of these data is currently lacking.

Such analysis would invariably be able to provide insightful information on global epidemiology, geographical distribution, and gender difference in prevalence of aerophagia in children which would be valuable for both clinicians and healthcare policy makers. With these objectives in mind, we conducted a systematic review and a meta-analysis of the epidemiology of AP in children.

## Methods

### Literature search and study selection

A literature search was conducted (by CK) using MEDLINE (1910 to March 2020), EMBASE (1947 to March 2020), PsycINFO (up to March 2020) and Web of Science (1900 up to March 2020) to identify studies reporting prevalence of AP. We set the age limit as from birth to 18 years. The search strategy used the following terms; Aerophagia [Text word] OR air swallowing [Text word] combined with epidemiology [Text word] OR epidemiologic study [Text word] OR prevalence [Text word] OR frequency [Text word]. Details of the search strategy are given in S1 Appendix.

There was no language restriction. AP was diagnosed based on any of the Rome criteria for children (Rome II, III, IV) [8–10]. Predetermined, eligibility criteria for inclusion of the studies are given below.

1. Studies including children 0–18 years

2. School or community-based studies

3. Defining aerophagia using the Rome criteria

4. Sample size more than 100

5. Reported prevalence of aerophagia

6. Published as a full paper

All abstracts identified after removal of duplications were screened for eligibility by two of the authors (SR, NMD). Once the irrelevant titles were excluded, all the potentially eligible manuscripts were read in detail to obtain the necessary particulars by the same authors (SR, NMD). A recursive search of the literature was also conducted using the bibliographies of all the eligible studies [11]. Disagreements were resolved by discussion.

## Quality assessment of selected studies

We conducted a quality assessment (SR and NMD) of all the eligible papers using a tool developed by Korterink et al. [12]. According to the tool, we evaluated all the selected manuscripts using the following six questions;

1. Is the method of subject selection described and appropriate?

2. Are subject characteristics sufficiently described

3. Is AP diagnosed with a Rome criterion?

4. Are the survey instruments reliable and valid?

5. Are the analytical methods described, justified and appropriate?

6. Were the results reported in sufficient details?

A 3-point scale was used to score each question (No [0], partial [1], Yes [2]). Higher scores indicate better methodological quality of the study. However, the score obtained for the quality assessment did not determine the inclusion or exclusion of the study into the systematic review and meta-analysis.

## Data extraction

SR and NMD extracted data from the eligible papers using Microsoft Excel spreadsheet (XP for professional edition; Microsoft, Redmond, WA). Yet again, we resolved discrepancies comparing and discussing the data set with the original paper. We extracted the following data for each individual study:

- name of the first author, year of publication,

- country of origin of the manuscript,

- population studied,

- the age range of the study sample,

- sample size,

- questionnaire used for the study,

- diagnostic criteria for AP,

- total prevalence,

- age-specific and sex-specific prevalence.

## Data synthesis and statistical analysis

Meta-analysis was performed using MedCalc for Windows, version 19.2.6 (MedCalc Software, Ostend, Belgium), and forest plots were generated using the same package. The heterogeneity of included studies was assessed with the Cochrane-Q-statistic and $I^2$ tests. A p value of 0.05 was used as the cutoff value for statistical significance. A P value $< .10$ and $I^2 > 50\%$ were considered significant heterogeneity. Pooled prevalence rates were calculated using a fixed-effect model in case of no significant heterogeneity; otherwise, the random-effect model was applied. Publication bias was evaluated by funnel plot and Egger tests; a P-value of $<0.05$ was considered statistically significant. However, expecting a significant heterogeneity among studies it

was decided not use a cutoff value to exclude studies from the meta-analysis. We mapped the country-specific estimated prevalence (obtained from the meta-analysis) in the world map using ArcGIS 10.2, and ESRI base map/base map outline (Esri, Redlands CA).

# Results

## Literature search

Our search criteria identified 76 titles. After the removal of duplications, 57 titles were screened for compliance with the strict eligibility criteria. Twenty-two (22) full-text papers were reviewed in-depth, out of which three studies were found to be hospital-based and were excluded. The process left us with 19 relevant studies [1–3, 6, 7, 13–26] Fig 1 shows the PRISMA diagram for the study. Table 1 depicts the details of all studies included in the systematic review and the meta-analysis.

## Characteristics of studies

All 19 studies were cross-sectional studies from various parts of the world. All were school-based surveys. There were nine data sets from seven Asian countries [1–3, 13–16], two studies from nine European countries [16, 17], two studies from North America, both from USA [7, 18], two studies from central America [19, 20] and seven studies from five South American countries [6, 21–26]. A study from Sri Lanka has given the prevalence of AP according to two iterations of Rome criteria (Rome II and Rome III) [13]. We selected the prevalence value from the Rome III criteria for the analysis. All studies except three have used Rome III criteria to diagnose AP [6, 7, 14].

## Quality assessment

Table 2 shows the quality assessment of all 14 studies. All studies have used an iteration of Rome criteria (Rome II, III, or IV). Most of the studies scored full marks for the description of the target population, reliability of the data collection instrument, and the description of the analytical method. However, the quality of reporting results was partial in most of the studies.

## Criteria for quality assessment

1. Is method of subject selection described and appropriate?

2. Are subject characteristics sufficiently described?

3. Is aerophagia diagnosed appropriately?

4. Are the survey instruments reliable and valid?

5. Are the analytic methods described/justified and appropriate?

6. Were the results reported in sufficient details?

## Pooled prevalence of AP

The pooled prevalence of AP in all studies with a total of 40129 children and adolescents is 3.66 (95% confidence interval (CI) 2.44–5.12). The lowest prevalence was reported in Mexico [19] while the highest was found in Sri Lanka [1]. There was significant heterogeneity between studies [$I^2$ 98.06% with 95% CI 97.70–98.37) but no evidence of funnel plot asymmetry (Egger

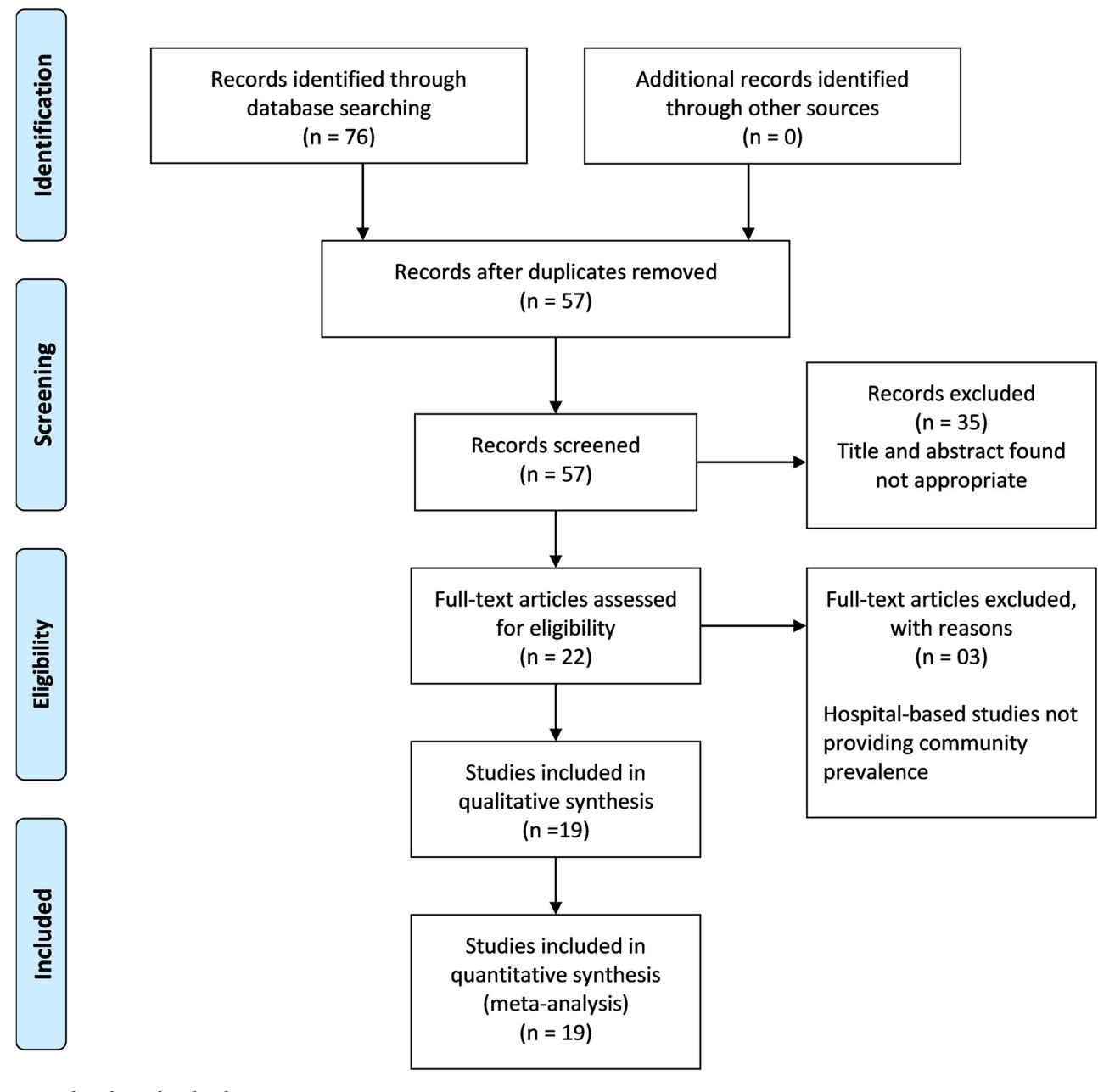

**Fig 1. Flow chart of study selection.**

test, P = 0.56). Fig 2 depicts the forest plot of all the epidemiological studies, and Fig 3 illustrates the global prevalence in the world map. Table 3 shows the pooled prevalence of AP according to the geographical locations. Three studies provide the gender-specific prevalence of AP [2, 3, 16]. When analyzed using the random effect model, the odds ratio (OR) for the males was 0.899 (95%CI 0.49–1.65), with $I^2$ value of 77.69 indicating gender does not affect the prevalence of AP.

One study reported the age-specific prevalence of AP [16]. According to their data, AP is more prevalent in the age group 11–18 years compared to 4–10 years. The age groups studied

**Table 1. Characteristics of the selected studies.**

| Name and the reference | Location | Population | Age range in years | Sample size | Case definition | Prevalence (%) |
|---|---|---|---|---|---|---|
| **Asia** | | | | | | |
| Sohrabi et al. (2010) [14] | Iran | School children | 14–19 | 1436 | Rome II | 3.3 |
| Devanarayana et al. (2011) [13] | Sri Lanka | School children | 10–16 | 427 | Rome III | 6.3 |
| Devanarayana et al. (2012) [3] | Sri Lanka | School children | 10–16 | 2163 | Rome III | 7.5 |
| Sagawa et al. (2013) [2] | Japan | School children | 10–17 | 3976 | Rome III | 2.0 |
| Bhatia et al. (2016) [15] | India | School children | 10–17 | 1115 | Rome III | 0.4 |
| Rajindrajith et al. (2018) [1] | Sri Lanka | School children | 13–18 | 2453 | Rome III | 15.1 |
| Scarpato et al. (2018) [16] | Israel | School children | 4–18 | 1222 | Rome III | 6.0* |
| Scarpato et al. (2018) [16] | Jordan | School Children | 4–18 | 1594 | Rome III | 7.3* |
| Scarpato et al. (2018) [16] | Lebanon | School Children | 4–18 | 1007 | Rome III | 4.4* |
| **Europe** | | | | | | |
| Bouzios et al. (2017) [17] | Greece | School children | 6–17 | 1588 | Rome III | 3.5 |
| Scarpato et al. (2018) [16] | Croatia | School children | 4–18 | 1716 | Rome III | 18.3* |
| | Greece | School children | 4–18 | 1316 | Rome III | 6.3* |
| Scarpato et al. (2018) [16] | Italy | School children | 4–18 | 2118 | Rome III | 2.6* |
| Scarpato et al. (2018) [16] | Macedonia | School children | 4–18 | 1555 | Rome III | 6.0* |
| Scarpato et al. (2018) [16] | Serbia | School children | 4–18 | 1657 | Rome III | 2.9* |
| Scarpato et al. (2018) [16] | Spain | School children | 4–18 | 1565 | Rome III | 2.9* |
| **USA** | | | | | | |
| Lewis et al. (2016) [18] | USA | School children | 4–18 | 949 | Rome III | 4.3 |
| Robin et al. (2018) [7] | USA | School children | 8–14 | 959 | Rome IV | 0.3 |
| **Central America** | | | | | | |
| Dhroove et al. (2017) [19] | Mexico | School children | 8–18 | 362 | Rome III | 0.0 |
| Lu et al. (2016) [20] | Panama | School children | 8–14 | 321 | Rome III | 0.3 |
| **South America** | | | | | | |
| Zablah et al. (2015) [26] | El Salvador | School children | 8–15 | 399 | Rome III | 0.5 |
| Jaime et al. (2018) [22] | Chile | School children | 7–19 | 506 | Rome III | 13.4 |
| Jativa et al. (2016) [23] | Ecuador | School children | 8–15 | 417 | Rome III | 2.6 |
| Nelissen et al. (2018) [21] | Argentina | School children | 12–18 | 483 | Rome III | 5.6 |
| Saps et al. (2017) [24] | Colombia | School children | 8–18 | 4394 | Rome III | 0.8 |
| Saps et al. (2018) [6] | Colombia | School children | 8–18 | 3567 | Rome IV | 0.5 |
| Peralta-Palmezano et al. (2019) [25] | Colombia | School children | 8–17 | 864 | Rome III | 0.1 |

*Prevalence of children between 11–18 years

even varied among studies carried out by the same research groups. In Sri Lanka, two studies have used the age group 10–16 years, whereas the other study by the same group has recruited children of 13–18 years [1, 3, 13]. Similarly, studies from South America have recruited varying age groups in their studies [23–25].

## Discussion

This systematic review and meta-analysis assembled all the population-based studies in children to compute the global epidemiology of AP. The pooled prevalence of AP was 3.66%. There was no gender difference in the prevalence of AP. The pooled prevalence was highest at Asia while the lowest was noted in the Central America.

AP is a clinical condition prevalent across the world, which is characterized by repetitive swallowing of air, abdominal distension, and passing the swallowed air either as burps or

Table 2. Quality assessment of studies.

| Study | 1 | 2 | 3 | 4 | 5 | 6 | Total |
|---|---|---|---|---|---|---|---|
| Bhatia et al. 2016 [15] | 2 | 1 | 2 | 2 | 2 | 1 | 10 |
| Bouzios et al. 2017 [17] | 2 | 2 | 2 | 2 | 2 | 1 | 11 |
| Devanarayana et al. 2011 [13] | 2 | 2 | 2 | 2 | 2 | 1 | 11 |
| Devanarayana et al. 2012 [3] | 2 | 2 | 2 | 2 | 2 | 2 | 12 |
| Dhroove et al. 2017 [19] | 1 | 2 | 2 | 2 | 2 | 1 | 10 |
| Jaime et al. 2018 [22] | 2 | 2 | 2 | 2 | 2 | 1 | 11 |
| Jativa et al. 2016 [23] | 1 | 2 | 2 | 2 | 2 | 1 | 10 |
| Lewis et al. 2016 [18] | 1 | 2 | 2 | 2 | 2 | 1 | 10 |
| Lu et al. 2016 [20] | 1 | 2 | 2 | 2 | 2 | 1 | 10 |
| Nelissen et al. 2018 [21] | 1 | 2 | 2 | 2 | 2 | 1 | 10 |
| Peralta-Palmezano et al. 2019 [25] | 2 | 2 | 2 | 2 | 2 | 1 | 11 |
| Rajindrajith et al. 2018 [1] | 2 | 2 | 2 | 2 | 2 | 2 | 12 |
| Robin et al. 2018 [7] | 1 | 2 | 2 | 2 | 2 | 1 | 10 |
| Sagawa et al. 2013 [2] | 1 | 2 | 2 | 2 | 2 | 2 | 11 |
| Saps et al. 2017 [24] | 1 | 2 | 2 | 2 | 1 | 1 | 09 |
| Saps et al. 2018 [6] | 1 | 2 | 2 | 2 | 2 | 1 | 10 |
| Scarpato et al. 2018 [16] | 2 | 2 | 2 | 2 | 2 | 2 | 12 |
| Sohrabi et al. 2010 [14] | 2 | 2 | 2 | 2 | 2 | 1 | 11 |
| Zablah et al. 2015 [26] | 1 | 2 | 2 | 2 | 2 | 1 | 10 |

No; 0 points, Partial;1 point, Yes; 2 points

flatus. In the present analysis, the pooled prevalence was noted to be 3.66% across all studies. The reported prevalence ranged from 0.0% in Mexico to 15.1% in Sri Lanka [1, 19]. The pooled prevalence value was much closer to the prevalence in the US and Europe [17, 18] and some Asian studies [14], but higher than most of the studies from Central and South America [6, 19,

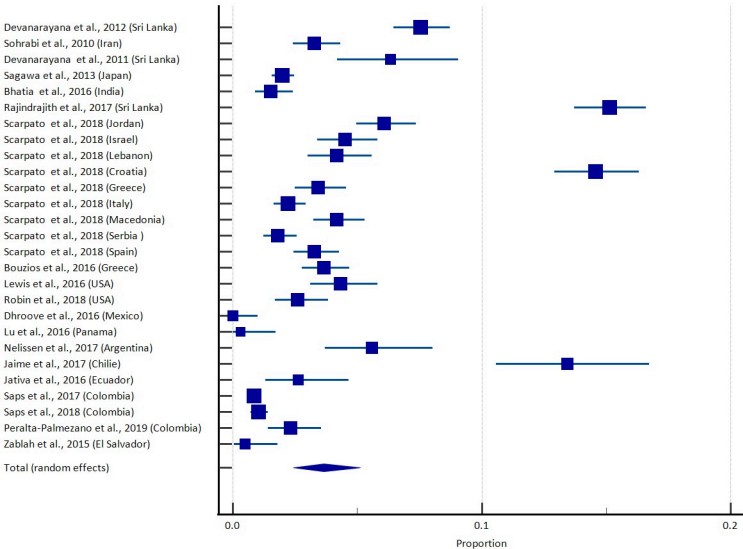

Fig 2. Forest plot prevalence of aerophagia.

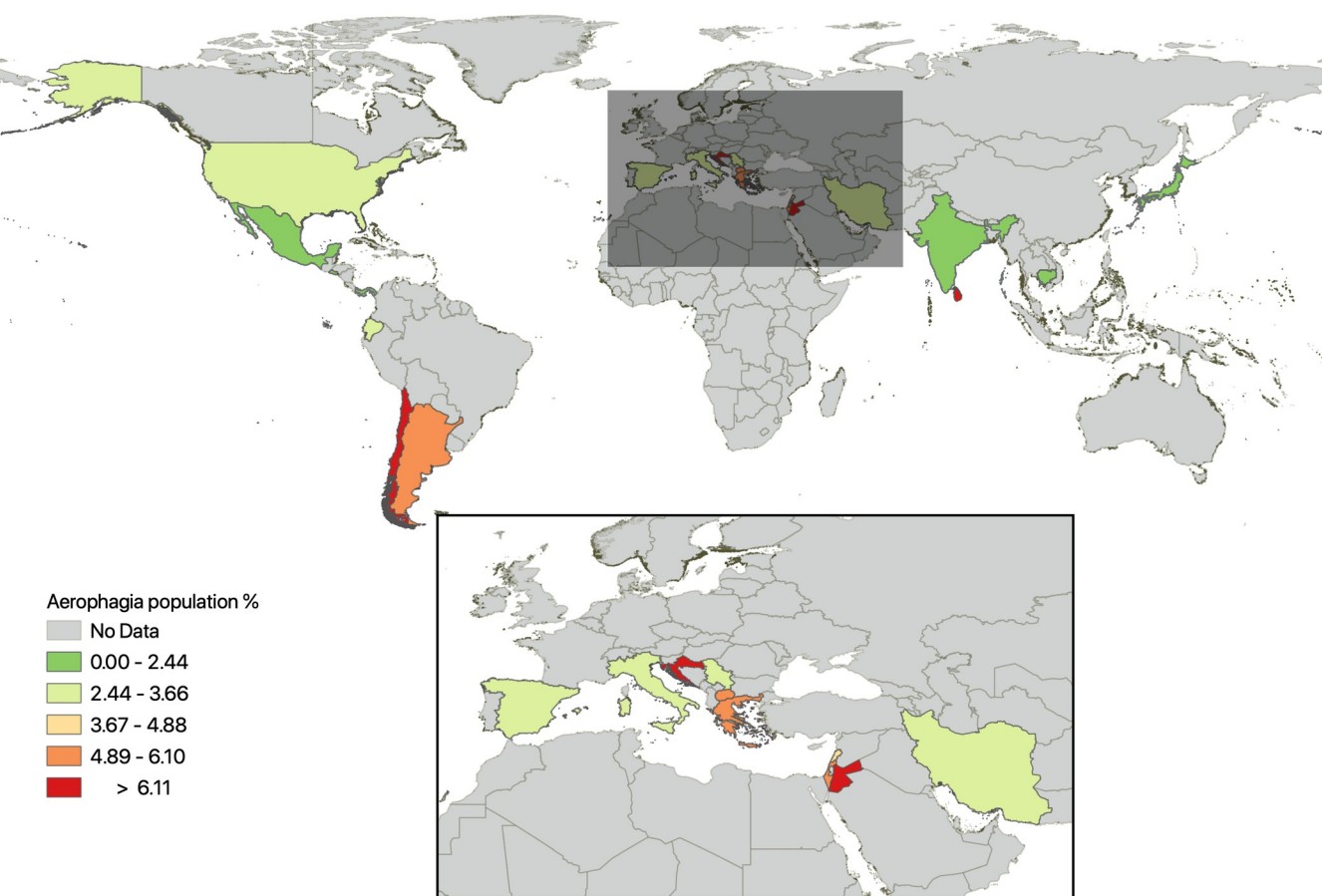

**Fig 3. Prevalence of aerophagia: The world map.** Final map was created using ArcGIS software by ESRI, using Basemaps supported by Esri under a license, original Copyright 2019 ESRI. All rights reserved.

20, 25, 26]. We noted that there is a wide variation in the prevalence from country to country and continent to continent. The pathophysiology of AP is related to air swallowing, esophageal motility and supragastric belching. It is unlikely that these factors change drastically between countries. One of the possibilities is the lack of uniformity in translating the Rome III questionnaire to different languages across the world. There could be subtleties in the meaning of key symptoms during the translation of the questionnaire, which may affect the reporting of symptoms.

Furthermore, the cultural and linguistic differences in the interpretation of symptoms such as air swallowing, belching, and flatus may be different from country to country. Variation in

**Table 3. Pooled prevalence of aerophagia related to geographical location.**

| Geographical location | Studies | Subjects | Pooled prevalence | 95% Confidence interval |
|---|---|---|---|---|
| Asia | 9 | 15393 | 5.13 | 2.69–8.29 |
| Europe | 2 | 11515 | 4.21 | 1.98–7.20 |
| North America | 2 | 1908 | 3.46 | 1.98–5.33 |
| Central America | 2 | 683 | 0.21 | 0.01–0.91 |
| South America | 7 | 10360 | 2.92 | 1.25–5.27 |

consumption of food items and differences in feeding practices in children also could have contributed to the differences. The other potential reasons for the variability include ethnic diversity and genetic variations. The differences in survey methods (internet surveys, school survey questionnaires filled by adolescents, questionnaires filled by parents at home etc.) may also have played a possible role in differences in the prevalence.

Only a few studies have provided an in-depth analysis of basic parameters such as age and gender-related prevalence. One study from Sri Lanka and a study from Japan provide data on gender-related prevalence. According to the meta-analysis, there is no difference in AP among boys and girls. Age-related prevalence of nine European countries is reported by Scarpato *et al* in their survey of pediatric functional gastrointestinal disorders in the Mediterranean region [27]. The data are only for two groups (4–10 years and 11–18 years), not adequately descriptive enough for a meta-analysis. However, the general trend across the included studies is that the prevalence increases with age.

Although it appears to be simple air swallowing, belching, and flatulence, all of which are harmless symptoms, AP is known to affect negatively to the lives of affected children [1]. Two studies have shown that children with AP are suffering from a multitude of somatic symptoms, psychological maladjustments, poor academic performances, and poor health related quality of life [1–3]. In this sense it is important to understand the epidemiology at a global level to improve healthcare of children with AP by implementing awareness programs and developing strategies to allocate healthcare resources.

There are several strengths of this systematic review and meta-analysis. The total number of children included in the epidemiological surveys was over 40,000 giving the facility of large number of children to draw conclusions. All studies were conducted over an eight-year period (2010–2018), and all studies except three used well defined Rome III criteria to diagnose AP, giving a much-needed uniformity for studies. The other two studies also used different iterations of Rome criteria. The effect size of the studies using Rome II and IV criteria is small and would not have affected the overall results in a skewed manner [7, 14]. We only included school-based surveys that represent general childhood population of the country. When assessed as to the quality of the selected studies, the majority of them were of high-quality providing reassurance of the robustness of our findings. Finally, we used the random effect model in the statistical calculations as in previous studies, which provides more conservative estimates [11].

Our study has several limitations as well. The assessed heterogeneity of the studies was significantly high with a $I^2$ value of 98.16. Differences in demographic characteristics of the recruited children, differences in ethnicity and cultures, and variation in the definition (only in 3 studies) could have contributed to this observation. Studies from several continents such as Africa and Australia were not available, leading to difficulty in calculating the true global prevalence. Although the study conducted by Scarpato and co-workers had data from nine countries, we could not include all nine countries into the meta-analysis separately [16]. Most studies have not included gender-specific prevalence and age-specific prevalence, and therefore we could not conduct meta-analyses on these essential aspects.

Our findings have several implications to shape future research on AP. Firstly, researchers need to be encouraged to study epidemiology in the other parts of the world to improve the precision of the global prevalence. In addition, the current study highlighted the deficiencies of existing research which will improve the quality of epidemiological research on AP. For an example, most studies failed to report age and sex specific prevalence rates. With our findings of world-wide prevalence of 3.66% and previously reported effects of AP on lives of children, clinicians and researchers are urged to investigate pathophysiological mechanisms such as supragastric belching and novel therapeutic options for this disorder.

In conclusion, this systematic review and meta-analysis reports the global pooled prevalence of AP as 3.66% with significant heterogeneity between studies. We were unable to report the exact gender and age-specific prevalence, due to lack of reporting in most of the studies. Understanding the epidemiological dynamics would invariably lead to clarity of the prevalence, risk factors, and effects that could be used to plan preventive strategies and resource allocation to minimize the suffering of children with AP.

## Supporting information

**S1 Checklist. PRISMA checklist.**
(DOC)

**S1 Appendix. Search strategies.**
(DOCX)

## Author Contributions

**Conceptualization:** Shaman Rajindrajith, Samath D. Dharmaratne, Niranga M. Devanarayana.

**Data curation:** Shaman Rajindrajith, Chandrani Kuruppu, Niranga M. Devanarayana.

**Formal analysis:** Damitha Gunawardane, Samath D. Dharmaratne.

**Investigation:** Chandrani Kuruppu.

**Methodology:** Shaman Rajindrajith, Damitha Gunawardane, Chandrani Kuruppu, Samath D. Dharmaratne, Nipul K. Gunawardena, Niranga M. Devanarayana.

**Software:** Damitha Gunawardane, Chandrani Kuruppu, Nipul K. Gunawardena.

**Supervision:** Shaman Rajindrajith.

**Validation:** Nipul K. Gunawardena, Niranga M. Devanarayana.

**Visualization:** Nipul K. Gunawardena.

**Writing – original draft:** Shaman Rajindrajith.

**Writing – review & editing:** Shaman Rajindrajith, Damitha Gunawardane, Chandrani Kuruppu, Samath D. Dharmaratne, Nipul K. Gunawardena, Niranga M. Devanarayana.

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
