## [Decision Letter · Decision Letter 0]

1 Dec 2021

PONE-D-20-23688Global epidemiology of aerophagia in children and adolescents: A systematic review and meta-analysisPLOS ONE

Dear Dr. Rajindrajith,

Thank you for submitting your manuscript to PLOS ONE. After careful consideration, we feel that it has merit but does not fully meet PLOS ONE’s publication criteria as it currently stands. Therefore, we invite you to submit a revised version of the manuscript that addresses the points raised during the review process.

Two reviewers have evaluated your manuscript and identified several aspects that need clarification and revision in order to meet PLOS ONE's publication criteria. Please pay particular attention to Reviewer 1's concerns and queries regarding your study design and search criteria.

We look forward to receiving your revised manuscript.

Kind regards,

Jamie Males

Staff Editor

PLOS ONE

Journal Requirements:

2. We note that Figure 3 in your submission contain map images which may be copyrighted. All PLOS content is published under the Creative Commons Attribution License (CC BY 4.0), which means that the manuscript, images, and Supporting Information files will be freely available online, and any third party is permitted to access, download, copy, distribute, and use these materials in any way, even commercially, with proper attribution. For these reasons, we cannot publish previously copyrighted maps or satellite images created using proprietary data, such as Google software (Google Maps, Street View, and Earth). For more information, see our copyright guidelines: http://journals.plos.org/plosone/s/licenses-and-copyright.

a. You may seek permission from the original copyright holder of Figure 3 to publish the content specifically under the CC BY 4.0 license.  

3. In the methods, please describe how to provide the results of the publication bias analysisi in the figures and state the specific test (Begg's or Egger's) used in the Methods section. Please also state the cut-off used to indicate heterogeneity using the I2 statistic in the Methods section.

Reviewers' comments:

Reviewer's Responses to Questions

**Comments to the Author**

1. Is the manuscript technically sound, and do the data support the conclusions?

Reviewer #1: Partly

Reviewer #2: Yes

2. Has the statistical analysis been performed appropriately and rigorously? 

Reviewer #1: I Don't Know

Reviewer #2: Yes

3. Have the authors made all data underlying the findings in their manuscript fully available?

Reviewer #1: Yes

Reviewer #2: Yes

4. Is the manuscript presented in an intelligible fashion and written in standard English?

Reviewer #1: Yes

Reviewer #2: Yes

5. Review Comments to the Author

Reviewer #1: This manuscript is based on a systematic review and meta analysis of studies reporting aerophagia in children and adolescents. The systematic review appears to have been well conducted. I have a major concern regarding the manuscript and I have found several minor inaccuracies.

Major concern -

it is not appropriate to undertake a systematic review and meta analysis of the studies reporting the prevalence of aerophagia and to include data on risk factors and consequences. Studies on prevalence need samples representative of a defined population but representativeness is not an essential requirement to look at risk factors or consequences. It is likely that there is more literature reporting data on risk factors and consequences but these would not have met the inclusion criteria used in this review. In the manuscript there had been no synthesis of data related to the risk factors for aerophagia or its influence on the lives of children. Suggest confining the manuscript to the prevalence of aerophagia

Minor issues -

1. I think "Global epidemiology" is rather misleading in the title for a review of the prevalence of a condition.

2. Israel, Jordan, and Lebanon are listed under Europe in Table 1 but all three are in Asia.

3. Reference number 27 is stated as the source of some of the data in Table 1, but this data has been extracted from reference number 16.

4. Line 187 "There are five studies from four Asian countries[1-3, 13-15]' ...". This not correct.

There are six references (1, 2, 3, 13, 14, & 15) and in Table 1 under Asia six studies from four countries have been listed.

5. Line 196 in the manuscript mentions 14 studies but Table 2 has 19 studies.

6. How was the data for figure 3 obtained? Is figure 3 really necessary?

Reviewer #2: Although aerophagia in children is clinically less important comparing to other medical conditions, underlying pathophysiology can be important and needs further investigation. This study describes prevalence of aerophagia in children across the world, and discuss its effects on their life and possible underlying causes, such as stress and maltreatments. This study sends a message that aerophagia in children might have underlying causes and it should be investigated accordingly to prevent children from unnecessary stress and/or maltreatments.

I would suggest a further research to assess super-gastric belching in children. Because supra-gastric belching in children has not been well documented yet, and it can also be related to stress factors.

6. PLOS authors have the option to publish the peer review history of their article (what does this mean?). If published, this will include your full peer review and any attached files.

Reviewer #1: **Yes: **Arunasalam Pathmeswaran

Reviewer #2: No

---

## [Author Response · Author response to Decision Letter 0]

20 Feb 2022

Answers to Editorial and Reviewer Comments

Editorial Comments

Comment

Response

All are in accordance with PLOS ONE’s style requirements

Comment 

We note that Figure 3 in your submission contain map images which may be copyrighted. All PLOS content is published under the Creative Commons Attribution License (CC BY 4.0), which means that the manuscript, images, and Supporting Information files will be freely available online, and any third party is permitted to access, download, copy, distribute, and use these materials in any way, even commercially, with proper attribution. For these reasons, we cannot publish previously copyrighted maps or satellite images created using proprietary data, such as Google software (Google Maps, Street View, and Earth). For more information, see our copyright guidelines: http://journals.plos.org/plosone/s/licenses-and-copyright.

Response

This figure was generated by the authors using the software ArcGIS 10.2 (Esri, Redlands, Canada). We used ESRI base map/base map outline. We have access to this software as it is a free access software. Therefore, our figures are original and do not need to obtain copyright permission from other authors, journals, or authorities. The caption of the figure 3 was updated according to the ESRI guidelines as well.

Comment

In the methods, please describe how to provide the results of the publication bias analysis in the figures and state the specific test (Begg's or Egger's) used in the Methods section. Please also state the cut-off used to indicate heterogeneity using the I2 statistic in the Methods section.

Response

The necessary details were included in the method section as suggested by the editor.

Comment

In your Data Availability statement, you have not specified where the minimal data set underlying the results described in your manuscript can be found. PLOS defines a study's minimal data set as the underlying data used to reach the conclusions drawn in the manuscript and any additional data required to replicate the reported study findings in their entirety. All PLOS journals require that the minimal data set be made fully available. For more information about our data policy, please see http://journals.plos.org/plosone/s/data-availability.

Response

This is a systematic review and a meta-analysis. All the data included in this study are available in the public domain as published full text articles. 

Comment 

Please include captions for your Supporting Information files at the end of your manuscript, and update any in-text citations to match accordingly. Please see our Supporting Information guidelines for more information: http://journals.plos.org/plosone/s/supporting-information.

Response

Included at the end of the manuscript.

Reviewer 1

This manuscript is based on a systematic review and meta analysis of studies reporting aerophagia in children and adolescents. The systematic review appears to have been well conducted. I have a major concern regarding the manuscript and I have found several minor inaccuracies.

Comment

it is not appropriate to undertake a systematic review and meta analysis of the studies reporting the prevalence of aerophagia and to include data on risk factors and consequences. Studies on prevalence need samples representative of a defined population but representativeness is not an essential requirement to look at risk factors or consequences. It is likely that there is more literature reporting data on risk factors and consequences but these would not have met the inclusion criteria used in this review. In the manuscript there had been no synthesis of data related to the risk factors for aerophagia or its influence on the lives of children. Suggest confining the manuscript to the prevalence of aerophagia.

Response

Agreeing with the reviewer, we removed the data on risk factor and limited our manuscript to prevalence of aerophagia

Comment

 I think "Global epidemiology" is rather misleading in the title for a review of the prevalence of a condition.

Response

We changed the title according to the reviewer comment.

Comment

Israel, Jordan, and Lebanon are listed under Europe in Table 1 but all three are in Asia.

Response

We thank the reviewer for pointing out this issue. It was corrected and Israel, Jordan, and Lebanon were included into Asian studies. 

Comment

Reference number 27 is stated as the source of some of the data in Table 1, but this data has been extracted from reference number 16.

Response

We thank the reviewer for pointing out this error. The data were extracted from reference 16. The error was corrected.

Comment

Line 187 "There are five studies from four Asian countries[1-3, 13-15]' ...". This not correct. 

Response

The studies were rearranged, and this error was corrected as pointed out by the reviewer.

Comment

There are six references (1, 2, 3, 13, 14, & 15) and in Table 1 under Asia six studies from four countries have been listed.

Response

This is correct. Three studies from Sri Lanka. The other countries were Iran, India, and Japan. Now we have added Jordan, Israel, and Lebanon to this list as suggested by the reviewer in his previous comment. 

Comment

Line 196 in the manuscript mentions 14 studies but Table 2 has 19 studies.

Response

We thank the reviewer for pointing out this error. It was corrected to 19.

Comment

How was the data for figure 3 obtained? Is figure 3 really necessary?

Response 

Each country had at least one prevalence rate for aerophagia. When there were more than one study providing prevalence, we calculated the average for the country through the meta-analysis. This was done for Sri Lanka and Colombia where we observed multiple prevalence rates. We believe that the prevalence map is necessary and if the reviewer is not disagreeing, we would like to keep it in the manuscript. 

Reviewer 2

Comment

Although aerophagia in children is clinically less important comparing to other medical conditions, underlying pathophysiology can be important and needs further investigation. This study describes prevalence of aerophagia in children across the world, and discuss its effects on their life and possible underlying causes, such as stress and maltreatments. This study sends a message that aerophagia in children might have underlying causes and it should be investigated accordingly to prevent children from unnecessary stress and/or maltreatments.

Response

Thank you for your comment

Comment 

I would suggest further research to assess super-gastric belching in children. Because supra-gastric belching in children has not been well documented yet, and it can also be related to stress factors.

Response

We have discussed the possibility of supra-gastric belching as an aetiological factor in the discussion stressing that it need to be looked at during evaluation.

---

## [Decision Letter · Decision Letter 1]

4 Jul 2022

Epidemiology of aerophagia in children and adolescents: A systematic review and meta-analysis

PONE-D-20-23688R1

Dear Dr. Rajindrajith,

We’re pleased to inform you that your manuscript has been judged scientifically suitable for publication and will be formally accepted for publication once it meets all outstanding technical requirements.

Kind regards,

Hugh Cowley

Staff Editor

PLOS ONE

Additional Editor Comments (optional):

Reviewers' comments:

Reviewer's Responses to Questions

**Comments to the Author**

1. If the authors have adequately addressed your comments raised in a previous round of review and you feel that this manuscript is now acceptable for publication, you may indicate that here to bypass the “Comments to the Author” section, enter your conflict of interest statement in the “Confidential to Editor” section, and submit your "Accept" recommendation.

Reviewer #2: All comments have been addressed

2. Is the manuscript technically sound, and do the data support the conclusions?

Reviewer #2: Yes

3. Has the statistical analysis been performed appropriately and rigorously? 

Reviewer #2: Yes

4. Have the authors made all data underlying the findings in their manuscript fully available?

Reviewer #2: Yes

5. Is the manuscript presented in an intelligible fashion and written in standard English?

Reviewer #2: Yes

6. Review Comments to the Author

Reviewer #2: Much improved by responding other reviewer's comments.

As I made comments on the previous occasion, this manuscript sends a message that aerophagia in children might have underlying clinical/psychological causes and it should be investigated accordingly. I would suggest a further study to assess supra-gastric belching in children.

7. PLOS authors have the option to publish the peer review history of their article (what does this mean?). If published, this will include your full peer review and any attached files.

Reviewer #2: No

---

## [Editor Report · Acceptance letter]

7 Jul 2022

PONE-D-20-23688R1 

Epidemiology of aerophagia in children and adolescents: A systematic review and meta-analysis 

Dear Dr. Rajindrajith:

I'm pleased to inform you that your manuscript has been deemed suitable for publication in PLOS ONE. Congratulations! Your manuscript is now with our production department. 

Kind regards, 

on behalf of

Mr Hugh Cowley 

Staff Editor

PLOS ONE